# Perceived Relational Empathy and Resilience in People with Spinal Cord Injury at the End of Acute Care: A Cross-Sectional Study

**DOI:** 10.3390/healthcare12161559

**Published:** 2024-08-06

**Authors:** Monika Zackova, Paola Rucci, Rossana Di Staso, Silvia Ceretti, Giuseppe Bonavina, Eric Delmestro

**Affiliations:** 1Montecatone Rehabilitation Institute, 40026 Imola, Italy; silvia.ceretti@montecatone.com (S.C.); giuseppe.bonavina@montecatone.com (G.B.); eric.delemestro@montecatone.com (E.D.); 2Department of Biomedical and Neuromotor Sciences, University of Bologna, 40126 Bologna, Italy; paola.rucci2@unibo.it (P.R.); rossana.distaso2@unibo.it (R.D.S.)

**Keywords:** patient-reported experience measures (PREMs), patient-reported outcome measures (PROMs), resilience, relational empathy, spinal cord injury, acute care

## Abstract

In patients with spinal cord injury (SCI), patient-reported outcomes (PROMs) and experience of care measures (PREMs) are extremely relevant for the prognosis. However, there is a paucity of research on these topics. We conducted a cross-sectional study to investigate the relationships between these patient outcomes and other demographic and clinical variables in adult SCI patients discharged from the intensive care unit of an Italian tertiary rehabilitation hospital. We administered the Consultation and Relational Empathy (CARE) for perceived relational empathy, the Spinal Cord Independence Measure III self-report (SCIM-SR) for functional autonomy, the Numeric Rating Scale (NRS) for pain, and the Connor–Davidson Resilience Scale (CD-RISC-10) for resilience. Study participants consisted of 148 adults with SCI; 82.4% were male, with a mean age of 49.9 years (SD = 16.6). The lesion was traumatic in 82.4% and complete in 74.3% of cases. The median length of hospital stays was 35 days (interquartile range—IQR = 23–60). Perceived relational empathy was positively associated with resilience (r = 0.229, *p* = 0.005) and negatively associated with the length of the stay and lesion completeness. Resilience had a weak negative association with pain (r = −0.173, *p* = 0.035) and was unrelated to other variables. Clinicians should consider the routine assessment of PREMs and PROMs in order to personalize post-discharge therapeutic plans and identify appropriate measures to ensure continuity of care.

## 1. Introduction

Over the past 10 years, health services in many EU countries have begun to collect indicators based on patient-reported information such as patient-reported experience measures (PREMs) [1,2] and patient-reported outcome measures (PROMs) [3,4] to assess the quality of care provided and for the accountability and performance evaluation of health systems. PROMs typically represent subjective outcomes that are relevant to patients because they measure the impact of healthcare on their usual activities and self-care. They include, for example, pain, perceived functional capacity, and self-efficacy in managing their disease. PREMs measure users’ experiences of care, such as being involved in treatment decisions, being treated with dignity and respect, and receiving information about available services; they provide a practical way to explore person-centered, integrated models of care. A systematic review has shown that improving patients’ experience of care is often associated with improvements in patient safety, clinical effectiveness, and health outcomes [5]. In people with spinal cord injury (SCI), where personal suffering adds to clinical complexity, perceived relational empathy and resilience are extremely relevant for the prognosis, as treatment experience may influence therapeutic outcomes [6,7,8]. As a matter of fact, the success of a rehabilitation project depends not only on the therapeutic approaches used but also, to a large extent, on the trustful relationship between healthcare professionals and the patient [9,10]. Thus, the assessment with tools measuring the subjective experience of care [11,12,13] should be added to traditional assessments of clinical outcomes such as pain and functioning.

However, there are currently few studies using PROMs, and even fewer using PREMS in people with SCI, and none in the intensive phase of hospitalization.

Among PREMs, the perceived relational empathy of healthcare professionals has received some attention in studies on people with SCI. In a U.S. study of patients receiving care from the U.S. Veterans Health Administration and SCI Model Systems, based on a survey and administrative databases, higher physical and mental health status and tetraplegia were associated with greater perceptions of holistic care and empathy in the therapeutic patient–provider relationship [14]. In a second study conducted in India in patients seen for a follow-up visit at least one year after injury and living in the community, perceived relational empathy was associated with higher levels of quality of life [15].

In addition, resilience, a psychological characteristic, has been reported to be central to the well-being of patients with SCI. Resilience has been considered as a state-like variable that involves behaviors, thoughts, and actions, which can be learned over time, or as a trait-like construct that someone either has or does not have.

Psychological resilience can be conceptualized as the ability to (1) resist the negative effects of stressors, (2) “bounce back” from stressors, and/or (3) grow from stressors [16]. Hence, assessing an individual’s state of resilience and approach to setbacks may provide the development of important interventions that may involve, for example, bolstering resilience through external support systems such as social support and good relationships. This kind of resilience-specific intervention could also be important in addressing rehospitalization and extending benefits after discharge as well, such as through the adoption of new resilient coping strategies [17,18,19].

According to recent research, high levels of resilience are linked to emotional regulation, cognitive coping (e.g., problem-solving skills, reappraisal of the meaning of setbacks), and positive mental health [20]. Resilience was also found to be consistently related to acceptance of disability and high self-efficacy and, in general, to a better and satisfactory quality of life [21,22,23,24,25].

To our knowledge, no research study has investigated the relationship between resilience and patient-perceived relational empathy of doctors at discharge from acute hospital care.

Our results have the potential to contribute to the future development of strategies to improve and personalize the care provided to patients with SCI.

The primary aim of this study is to describe the perceived relational empathy of doctors and the resilience of patients with SCI at discharge from the intensive care unit (ICU) of the Montecatone Rehabilitation Institute.

The secondary aims are to analyze the association of perceived relational empathy and resilience with pain intensity, functional autonomy achieved by the patient, and the medical history of patients at discharge.

## 2. Materials and Methods

This observational, cross-sectional study was carried out at the Montecatone Rehabilitation Institute and included people with SCI admitted to the ICU.

Patients in the ICU are mainly referred from the resuscitation services, emergency departments, and trauma centers of other Italian hospitals, usually shortly after the trauma. Referred patients have completed the surgical phase, are still clinically very unstable, and do not have the autonomy to perform basic vital functions. In the ICU, the post-traumatic situation is assessed, and the most appropriate measures are taken to start rehabilitation as early as possible and to ensure a high quality of care, guaranteeing the start of motor and respiratory rehabilitation at a very early stage, when respiratory, metabolic, nutritional, infectious, or other problems are still present and require intensive care, monitoring, and support. In this way, the ICU of the Institute initiates the process of true global rehabilitation for patients who cannot be weaned from the ventilator, thus becoming the link between the period of intensive intervention and the subsequent global rehabilitation phase, including training to promote patient and caregiver autonomy.

### 2.1. Study Population

Patients were enrolled in this study over a period of 37 months (June 2020–July 2023) and were assessed at discharge from the ICU after signing an informed consent to participate in the study and to have their personal data processed. For patients who were able to consent but were unable to sign, the consent form was signed by an impartial third party who was not involved in the study. The study was approved by the Ethics Committee CE AVEC protocol # 46-2021-OSS-AUSLIM.

Inclusion criteria were adult age, SCI at any neurological level, completeness and etiology, sufficient knowledge of Italian to complete questionnaires, and ability to provide written informed consent. Exclusion criteria were cognitive deficits, brain damage, and ongoing pregnancy.

The demographic variables and the characteristics of the lesion were extracted from electronic clinical records and linked with questionnaire data. These variables included age, gender, neurological level, AIS level (coded as A, complete lesion, and B, incomplete lesion), etiology (coded as traumatic or non-traumatic), comorbidity (coded as present or absent), and length of hospital stay.

### 2.2. PROMs

The Spinal Cord Independence Measure III self-report (SCIM-SR) [26] is a self-report questionnaire that assesses the individual’s autonomy by evaluating the ability to perform simple daily life tasks. It consists of 16 items organized into 3 groups: self-care (range 0–20), breathing and sphincter management (range 0–40), and mobility (range 0–40). Higher scores denote better functioning. A score of 0 indicates the need for total assistance in performing the task. The total score ranges from 0 to 100.

The Italian version of the SCIM-SR was validated by Bonavita et al. [27] against the interview-based version. A good agreement was found between the domain and total scores of the two versions (r = 0.918 for ‘Self-care’, 0.806 for ‘Respiration and sphincter management’, 0.906 for ‘Mobility’, and 0.934 for the total score).

The Numeric Rating Scale (NRS) measures the intensity of pain. It is a self-rated pain scale ranging from 0 to 10, where 0 indicates ‘no pain’ and 10 indicates the ‘worst pain imaginable’.

### 2.3. PREMs

The Consultation and Relational Empathy (CARE) measure is a patient-reported experience measure developed in the United Kingdom [11,28,29], which has been extensively validated [14,28,29,30,31,32] and shown to be highly reliable in differentiating between doctors [29,30,31]. It consists of ten items, rated as “poor” = 1, “fair” = 2, “good” = 3, “very good” = 4, “excellent” = 5, or “does not apply”. The total score is obtained as a sum of item scores and ranges from 10 to 50; higher scores reflect greater patient-centeredness, specifically patient perceptions of holistic care and relational empathy in their healthcare. This measure has been used in U.S. patients with spinal cord injury [14]. More information on CARE can be found in the CARE Measure Website [33]. In the Italian version [34], internal consistency was excellent (Cronbach’s α= 0.962), and the exploratory factor analysis confirmed the unidimensional structure of the CARE measure with 74.82% of variance explained by the first factor.

### 2.4. Resilience

The Connor–Davidson Resilience Scale (CD-RISC-10) is a self-report tool with 10 items scored on a five-point Likert scale from 0 (not at all true) to 4 (true almost all of the time). It measures resilience and the ability to cope with adversity and demonstrates good construct validity and internal consistency (α = 0.85) in an undergraduate sample [13]. Higher scores indicate higher levels of resilience. The total score ranges between 0 and 40. In the Italian version, validated by Di Fabio and Palazzeschi [35] in university students, the scale proved to be unidimensional in confirmatory factor analysis and showed good reliability (Cronbach’s alpha = 0.89) and concurrent validity.

### 2.5. Statistical Analyses

A sample size of 150 was calculated to detect a correlation ≥ 0.20 between resilience and relational empathy, with an 80% power and a type-I error of 0.05.

Continuous variables were summarized using the mean and standard deviation or the median and interquartile range, and categorical and ordinal variables were summarized using absolute and percentage frequencies. Correlations between continuous or ordinal variables were determined using Spearman’s correlation coefficient, and between a continuous and a dichotomous variable using the point-biserial correlation coefficient. Correlations between 0.10 and 0.29 were considered weak, between 0.30 and 0.49 moderate, and >0.50 strong, using Cohen’s criteria [36].

Comparisons between groups were performed using the *t*-test or Mann–Whitney’s test for continuous variables, and the χ^2^ test or Fisher’s exact test for categorical variables, as appropriate.

Multivariable stepwise linear regression was used to identify the variables associated with a better perception of relational empathy during healthcare encounters. Two continuous variables, the CD-RISC score and the length of hospital stay, and one categorical variable, the completeness of the lesion (coded as complete/incomplete), were included as independent variables in the model.

## 3. Results

This study sample included 148 patients (82.4% male, mean age of 49.9 years, SD = 16.6). The etiology of the lesion was traumatic in the majority of cases (82.4%) and the AIS level was A (complete lesion) in 74.3%. The median length of hospitalization was 35 days (range 7–738, IQR = 23–60) (Table 1).

Patients reported a minimal level of pain at discharge (median NRS = 1, IQR 0–3). Descriptive statistics of SCIM subscales and the total score, and of CARE and the CD-RISC scores are provided in Table 2.

The mean CARE score was 40.2 (SD = 7.4), a value located between the 10th and 25th percentile, i.e., in the lower quartile of normative scores, 39.9% of patients had a score exceeding the normative value of 43 (Figure 1).

The CD-RISC scale had a mean total score of 24.5 and a range from 0 to 40, showing that the participants exhibited significant variability, from very poor to excellent perceived resilience (Figure 2).

The frequency distribution of the CD-RISC items (Figure 3) indicated variability in responses according to the aspect of resilience investigated; higher scores were found in the ability to cope with adversities (item 2), not being easily discouraged by failure (item 8), and thinking about themselves as strong persons (item 9), while the ability to find the humorous side of the problem was very low (item 3).

The SCIM functional scores were indicative of the high severity of these patients’ conditions, with median scores of 0 for self-care and mobility, with 75% requiring total assistance with self-care, 45.3% with managing respiratory and sphincter function, and 92.6% with mobility.

### 3.1. Correlations between Perceived Relational Empathy, Resilience, Pain, and Functioning

The total CARE score was positively related to the CD-RISC score (r = 0.229, *p* = 0.005, Figure 4), indicating that higher perceived empathy in the relationship with the doctors was associated with higher resilience. Pain was negatively associated with resilience (r = −0.173, *p* < 0.05), suggesting that patients experiencing high levels of pain were less resilient. The SCIM scales were strongly associated with each other and with the total score of the scale, as expected.

### 3.2. Correlations of Perceived Relational Empathy Resilience, Pain, and Functioning with Age, Gender, Length of Stay, Etiology, and Neurological Level of Lesion

The level of perceived relational empathy with the doctors was negatively correlated with the length of stay and the completeness of the lesion, indicating that a longer stay and having a complete lesion were associated with poorer perceived empathy. Still, perceived empathy and pain were unrelated to demographic factors and the etiology and neurological level of the lesion (Table 3). The SCIM total score was negatively associated with the length of stay, tetraplegia, and lesion completeness, indicating that patients with better functioning had a shorter ICU stay and were less likely to have tetraplegia and a complete lesion. In addition, the total SCIM score was positively associated with traumatic etiology and comorbidity. Self-care and mobility scores were lower in males than in females.

When we performed a multivariable stepwise linear regression to identify the variables independently associated with a better perception of the relational empathy of doctors, only resilience (the CD-RISC score) emerged as a significant predictor and entered the model (b = 0.225, SE(b) = 0.081, *p* = 0.006).

## 4. Discussion

To our knowledge, few studies have used PREMs in this patient population [14,37,38,39,40,41], and this is the first study to provide information on PREMs, PROMs, and resilience. Our findings indicated that the ability of the doctors to build an empathic relationship with the patient was associated with resilience. Although no causal relationship was investigated in this cross-sectional study, it is reasonable to assume that relational empathy affects patients’ resilience and not vice versa. Notably, relational empathy was unrelated to functional outcomes and pain. However, we found a weak negative relationship between relational empathy and the length of stay and completeness of the lesion, suggesting that it may be more challenging to establish a positive relationship between a patient and clinician when the clinical condition is more serious, and hospitalization is of a longer duration. A recent study using the CARE measure in Indian patients with SCI found that quality of life was closely associated with greater perceptions of holistic care and empathy in the therapeutic patient–provider relationship and concluded that the lack of coordination, poor quality of life, and limited communication between a patient and clinician may occur when the latter focuses only on treating the disease rather than treating the patient as a “whole person” [15]. In addition, it has been shown that a good nurse–patient relationship reduces the length of hospital stay and improves the quality and satisfaction of both [10].

Overall, resilience levels in our sample were lower than those of a recent study on patients with SCI participating in a survey (the mean and SD for CD-RISC-10 were M = 29.68, SD = 5.71). This result can be explained by the greater severity of the patients’ conditions in our study, assessed on discharge from intensive care. Resilience was negatively associated with pain, indicating that patients suffering from pain perceived themselves as less capable of coping with their condition. We did not find a significant association between resilience and functioning in SCI. This is consistent with a study carried out in axial spondylarthritis, showing that levels of resilience did not contribute to patients’ perception of their disease activities [42]. Possible explanations are that resilience does not only depend on an individual’s ability to cope with adversities, but also on the strength of interpersonal relationships, the resources available in the community and family circles that can be used to facilitate resilience, biological factors, and stressors, as well as an individual’s level of self-awareness [43,44,45,46]. However, another study conducted on patients with SCI during the COVID-19 pandemic [47] found that autonomy, in addition to psychological health and participation, were determinants of resilience.

One limitation of our study is the cross-sectional design that does not allow us to determine the presence and the direction of causal links among variables.

Patients were recruited from a tertiary hospital in which cases of SCI are more severe than those managed in the community. The mean resilience score for our sample was 24.2, which was slightly lower than that of other cohort studies and might limit the generalizability of the results.

Possible determinants of resilience such as religiousness, depression, life satisfaction, and social support were not collected, thus leaving an important source of variability unexplained.

## 5. Conclusions

In summary, our study demonstrated that resilience and the perceived relational empathy of doctors are related to each other, but not resilience and functioning. Therefore, the experience of a positive and empathic relationship with doctors is beneficial to the patients and may empower them during rehabilitation. Our results can be generalized only to patients in the initial phase of rehabilitation, shortly after the lesion. Hence, it would be useful for clinicians to collect these subjective measures in order to have a clearer picture of the quality of care provided and of the patients’ psychological strengths; this may be used to accelerate potential clinical improvement during the rehabilitation period and after the transition to community care.

## Figures and Tables

**Figure 1 healthcare-12-01559-f001:**
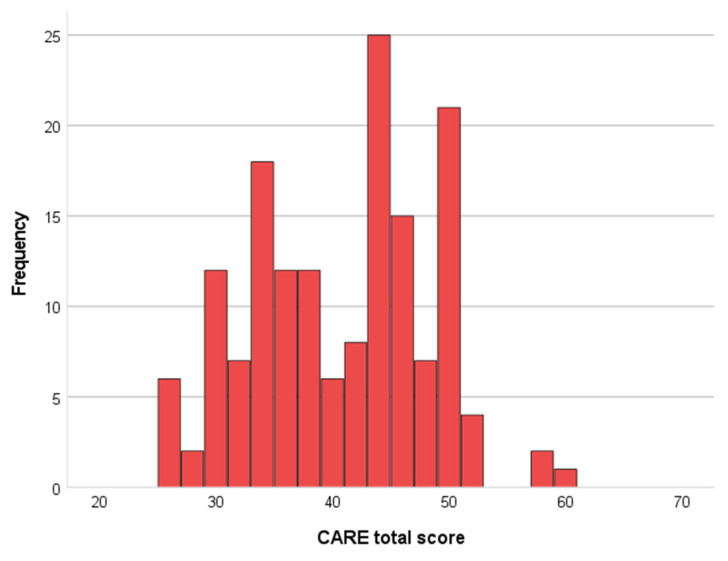
Histogram of the frequency distribution of CARE scores. Mean 40.47, SD = 7.496, N = 148.

**Figure 2 healthcare-12-01559-f002:**
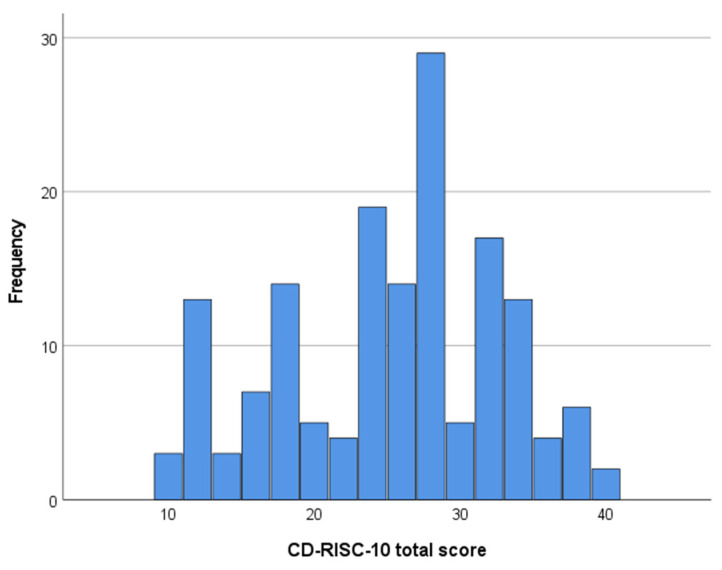
Frequency distribution of CD-RISC total score. Mean 24.9, SD = 7.53, N = 148.

**Figure 3 healthcare-12-01559-f003:**
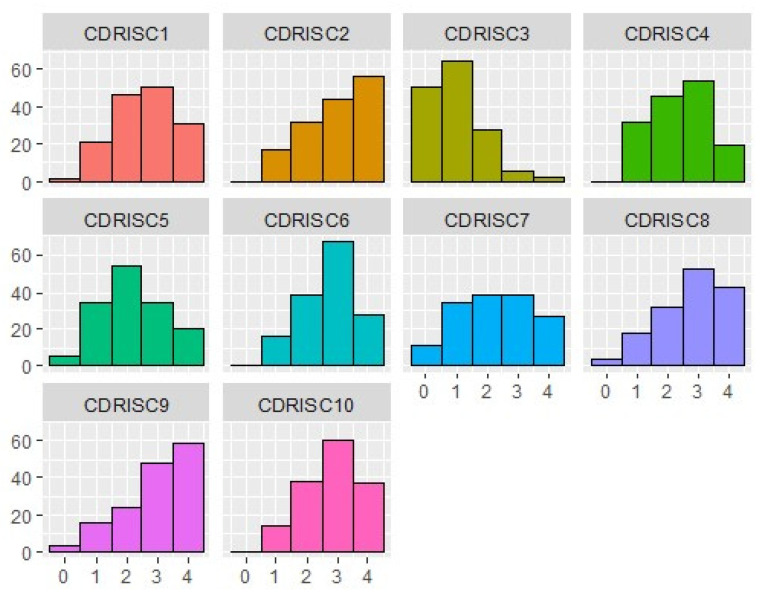
Frequency distribution of CD-RISC items. The x-axis reports the response options to the CD-RISC items on a 5-point Likert scale (0 = “not true”, 4 = “true nearly all the time”); the y-axis reports the percentage of responses.

**Figure 4 healthcare-12-01559-f004:**
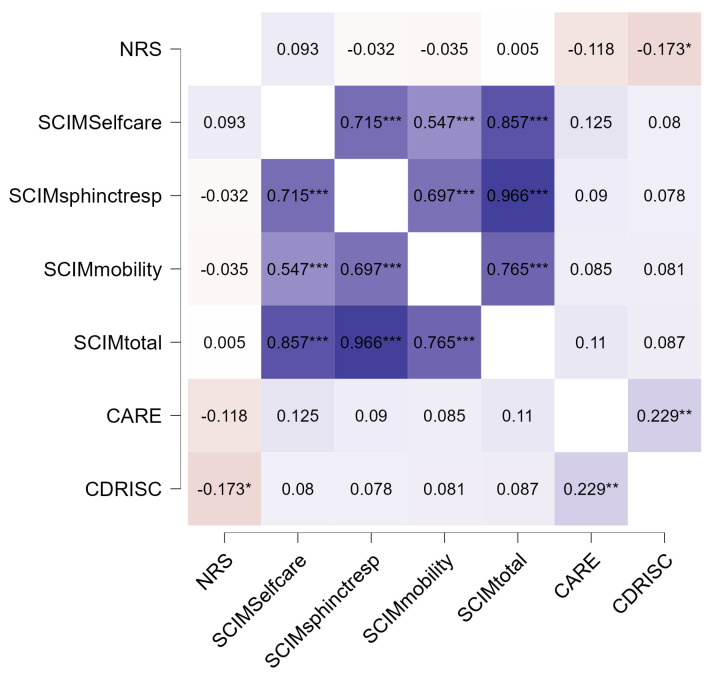
Heatmap showing the relationships of PROMs variables with perceived relational empathy and resilience. This heatmap uses a warm to cold color spectrum in which the warm areas’ values are high and the cold areas’ values are low. NRS, Numerical Rating Scale; SCIM, Spinal Cord Independence Measure; CARE, Consultation and Relational Empathy Measure; CD-RISC-10, 10-item Connor–Davidson Resilience Scale. * *p* < 0.05, ** *p* < 0.01, *** *p* < 0.001.

**Table 1 healthcare-12-01559-t001:** Characteristics of the sample (N = 148).

Variables	
Sex, *n* (%)	
M	122 (82.4)
F	26 (17.6)
Age, mean (SD)	49.9 (16.6)
Age, median (IQR)	52 (35.9–63.2)
Etiology, *n* (%)	
Traumatic	122 (82.4)
Non-traumatic	26 (17.6)
ASIA Impairment Scale (AIS), *n* (%)	
A	110 (74.3)
B	38 (25.7)
Length of hospitalization, mean (SD)	55.8 (83.8)
Length of hospitalization, median (IQR))	35 (23–60)

ASIA Impairment Scale (AIS).

**Table 2 healthcare-12-01559-t002:** Descriptive statistics of scaled variables.

Variables	
NRS ^1^ score, mean (SD)	1.52 (1.76)
NRS ^1^ score, median (IQR)	1 (0–3)
NRS classes, *n* (%)	
1–3	129 (87.2)
4–6	19 (12.8)
7–10	0
SCIM ^2^ Self-care scale, mean (SD)	1.09 (2.45)
SCIM ^2^ Self-care scale, median (IQR)	0 (0–0.75)
SCIM sphinct-respiratory scale, mean (SD)	3.05 (4.40)
SCIM sphinct-respiratory scale, median (IQR)	2 (0–4)
SCIM mobility scale, mean (SD)	0.20 (0.96)
SCIM mobility scale, median (IQR)	0 (0–0)
SCIM total score, mean (SD)	4.33 (6.96)
SCIM total score, median (IQR)	2 (0–6.75)
CARE ^3^ score, mean (SD)	40.15 (7.39)
CARE ^3^ score, median (IQR)	41 (34–46)
CD-RISC ^4^ score, mean (SD)	24.44 (7.42)
CD-RISC ^4^ score, median (IQR)	26 (17–30)

^1^ NRS, Numerical Rating Scale; ^2^ SCIM Spinal Cord Independence Measure; ^3^ CARE, Consultation and Relational Empathy Measure; ^4^ CD-RISC-10, 10-item Connor–Davidson Resilience Scale.

**Table 3 healthcare-12-01559-t003:** Correlation of PREM/PROM scores and resilience with demographic and clinical factors.

Variables	Gender	Age	Length of Stay	Traumatic Etiology	Tetraplegia	Comorbidity	Complete Lesion
CARE ^1^	−0.003	−0.066	−0.186 *	0.054	−0.057	−0.055	−0.176 *
CD-RISC ^2^	−0.082	−0.017	−0.132	−0.125	0.013	−0.098	−0.095
NRS ^3^	−0.056	0.076	0.041	−0.036	−0.135	0.039	−0.029
SCIM ^4^ Self-care	−0.211 *	0.041	−0.248 **	0.109	−0.617 **	0.201 *	−0.378 **
SCIM sphinct-resp	−0.067	0.064	−0.344 **	0.227 **	−0.592 **	0.259 **	−0.303 **
SCIM mobility	−0.212 **	−0.136	−0.091	0.068	−0.325 **	0.012	−0.251 **
SCIM total score	−0.090	0.080	−0.329 **	0.263 **	−0.598 **	0.273 **	−0.338 **

* Correlation is significant at the 0.05 level (2-tailed). ** Correlation is significant at the 0.01 level (2-tailed). ^1^ CARE, Consultation and Relational Empathy Measure; ^2^ CD-RISC-10, 10-item Connor–Davidson Resilience Scale; ^3^ NRS, Numerical Rating Scale; ^4^ SCIM, Spinal Cord Independence Measure.

## Data Availability

The raw data supporting the conclusions of this article will be made available by the authors upon request.

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
