# Peer review of "Perceived Relational Empathy and Resilience in People with Spinal Cord Injury at the End of Acute Care: A Cross-Sectional Study"

_healthcare, 2024, doi:10.3390/healthcare12161559_

Round 1
Reviewer 1 Report
Comments and Suggestions for Authors
I congratulate you for your interesting and innovative study.
I worked as a male nurse in a national hospital for spinal cord injuries for several years.
Some questions and comments:
1) Why do you use the median, instead of the mean and SD? (in abstract and results -line 163-)
2) In 2.3. PREMs: Were patients asked about their relationship with doctors, or with healthcare personnel in general? It is an important issue, because most of the care time and basic needs are covered by nursing staff.
It is also interesting to specify it in Discussion (line 247-249).
Reviewer 2 Report
Comments and Suggestions for Authors
Please find attached file.

Reviewer 3 Report
Comments and Suggestions for Authors
· In general, the paper is well-written, it addresses relational empathy and resilience in people with spinal cord injury, this is important because the emotional health could be overlooked in these critical cases
· Consistency Cronback’s alpha for all the scales should be provided given that they were administered in Italian, except for the pain NRS scale
· Please add exclusion criteria
· The methods are nicely written
· The sample size is better to be justified either by power calculations or by previously published articles
· The resilience is associated with many factors, for example, Italians are known for their religiousness this should be elaborated more in the discussion
· Please expand the conclusion, how these results can be generalized? If applicable?
· The tables are not formatted according to the journal’s guidelines
· A figure showing the associations would be easier for the readers to comprehend the theme of the manuscript
